# Egg Protein Compositions over Embryonic Development in *Haemaphysalis hystricis* Ticks

**DOI:** 10.3390/ani14233466

**Published:** 2024-11-30

**Authors:** Qiwu Tang, Tianyin Cheng, Wei Liu

**Affiliations:** College of Veterinary Medicine, Hunan Agricultural University, Changsha 410128, China; qiwutang@stu.hunau.edu.cn

**Keywords:** tick, egg proteins, embryonic development, LC-MS/MS, LC-PRM/MS

## Abstract

Ticks, as obligate ectoparasites, pose significant health risks by transmitting pathogens to various hosts, including humans and livestock, resulting in diseases and economic losses. This study focuses on *Haemaphysalis hystricis*, a tick prevalent in Southeast Asia, which infests both wild and domestic animals, leading to severe health impacts like emaciation and growth stunting. The tick’s life cycle, which includes stages from egg to adult, is crucial for understanding its development and for devising effective control strategies. This research aims to characterize protein expression in *H. hystricis* eggs throughout embryonic development using LC/MS/MS techniques. By identifying key proteins involved in egg development and metabolism, the study seeks to provide insights into tick embryogenesis and to identify potential targets for interventions that could disrupt tick reproduction and reduce the transmission of tick-borne diseases. Ultimately, the findings are expected to contribute to improving public health and animal husbandry practices by informing better tick management strategies.

## 1. Introduction

Ticks, obligate ectoparasites, infest a wide range of hosts, including mammals, birds, and reptiles, such as wild deer, rodents, and various bird species, along with domestic animals like dogs and livestock. Renowned for transmitting pathogenic microorganisms—such as protozoa (e.g., *Babesia* spp.), bacteria (e.g., *Borrelia*, *Rickettsia*, *Ehrlichia* spp.), and viruses—they cause significant health issues, including anemia, dermatitis, and the transmission of tick-borne diseases, which affect both human and animal health, leading to economic losses in livestock production. Effective tick population management and disease prevention are therefore essential to mitigating their impact on public health and animal husbandry.

*Haemaphysalis hystricis* [1], an Ixodidae family member, is widespread in the temperate and subtropical hilly and mountainous regions of Southeast Asia, including Japan [2], Malaysia [3], and Indonesia [4], and has also been recorded in several Chinese provinces such as Taiwan [5], Wuhan [6], and Sichuan [7]. This tick species predominantly parasitizes wild hosts like wild boars [3,8], hedgehogs, pangolins [9], and giant pandas [10], as well as domestic animals, including cattle, goats [6], and dogs [2,7]. Infestation can result in host emaciation and stunted growth, while ticks act as vectors for various pathogens, including *Trypanosoma* [11], *Ehrlichia*, *Rickettsia* [6,12], *Borrelia* [3], and *Babesia* [5,13].

The tick life cycle encompasses four stages: egg, larva, nymph, and adult. During the egg stage, fertilization initiates complex developmental processes—cleavage, blastulation, gastrulation, and segmentation—culminating in larval emergence. These intricately regulated stages are key to understanding embryogenesis and devising advanced tick control strategies.

A thorough characterization of egg protein expression throughout development is essential for elucidating tick developmental mechanisms. Previous studies have characterized protein profiles of tick salivary glands and saliva, midgut, midgut contents, and eggs [14,15,16,17,18]. Several proteins involved in egg development across different tick species have been identified, including nutrient-related proteins like yolk protein and vitellogenin (also functioning as a nutrient in eggs) and lipocalin; enzymes such as heme-binding aspartic proteinases; vitellin-degrading cysteine endopeptidases; cathepsin L-like proteinases [19,20,21]; protease inhibitors like serpin, cysteine, and Kunitz domain-containing proteins; and immune proteins including antimicrobial peptides (AMP), macroglobulin, and cysteine-rich proteins. Additionally, enzymes such as GST, which exhibit free radical scavenging activity [22]; serine protease inhibitors [23]; and antimicrobial peptides [24] have been identified. Investigating the egg development of *H. hystricis* can enhance the understanding of tick embryogenesis and aid in screening candidate molecules for developing novel interventions that disrupt tick reproduction and pathogen transmission. This study employs LC/MS/MS methods to identify protein components in the protoplasm of *H. hystricis* eggs and analyze the dynamic changes in protein expression during egg development.

## 2. Materials and Methods

### 2.1. Collection of Ticks and Eggs

Xiangxi County, located in western Hunan Province, China, spans latitudes 27°4′ N to 29°5′ N and longitudes 108°6′ E to 110°6′ E, and is predominantly mountainous and hilly. Ticks were collected from wild boars (*Arctonyx collaris*) that had been rescued by the local wildlife protection department and kept at a wild animal farm in Xiangxi. The sampling was conducted from July to October 2022, with ticks being carefully removed using tweezers. The collection process was approved by the Hunan Provincial Department of Animal Protection and supervised by the Hunan Animal Health and Usage Committee (No. 43321503), ensuring no harm to the animals.

Ticks were gathered by local residents from the wild boars and transported to the laboratory in ventilated plastic bottles. Identification was performed based on morphological characteristics [25] observed under a light microscope, with confirmation via molecular methods following the protocol by Ernieenor et al. [26]. Concurrently, the ticks were weighed, and 20 engorged *H. hystrici* females, each weighing approximately 350 mg, were selected. These ticks were placed individually in wells of a 6-well plate, which was then maintained in complete darkness at 28 °C and 85% relative humidity within an incubator.

Eggs laid by the ticks were collected and weighed daily at a consistent time until the ticks died. The collected eggs were pooled and divided into four aliquots of roughly 50 mg each. One aliquot underwent a dewaxing process using a 1.5 mL chloroform/methanol mixture (2:1) for 15–20 s [27], followed by vortexing and careful removal of the supernatant. Afterward, 1 mL of sterile deionized water was added, the sample was vortexed again for 15–20 s, and the supernatant was discarded. The eggs were then labeled with the date and promptly stored in liquid nitrogen.

The remaining three aliquots were incubated under identical conditions of darkness, temperature (28 °C), and humidity (85% RH). Eggs were collected on days 7, 14, and 21 of incubation, dewaxed, and stored in liquid nitrogen. However, the incubation period may vary depending on climate conditions in the field [28,29].

### 2.2. Egg Protein Extraction

For further processing, dewaxed eggs from days 0, 7, 14, and 21 were transferred to sterilized glass homogenizers. Each sample was combined with 0.3 mL of sterile saline and ground thoroughly. The homogenate was then transferred to 1.0 mL centrifuge tubes, with an additional 0.5 mL of sterile saline added. Samples were kept at 4 °C for 30 min, followed by centrifugation at 5000 rpm for 5 min. The supernatant was carefully collected for subsequent analyses.

Supernatants were mixed with lysis buffer containing 4% SDS, 0.1 M DTT, 150 mM Tris-HCl (pH 8.0), and 0.1 mL of a protease inhibitor cocktail (P9599, Sigma, St. Louis, MI, USA). The mixture was boiled for 3 min and sonicated on ice. The resulting crude extract was reheated in boiling water and centrifuged at 16,000× *g* for 10 min at 4 °C to clarify the solution. Protein concentration was measured using the BCA protein assay reagent (Bio-Rad, Hercules, CA, USA). Finally, the supernatants were stored at −80 °C for future use.

### 2.3. SDS-PAGE from Four Stages Egg

A 20 μg aliquot of egg protein supernatant from each sample was combined with 30 μL of SDT buffer (4% sodium dodecyl sulfate, 100 mM dithiothreitol, and 150 mM Tris-HCl, pH 8.0). The mixture was subjected to ultrasonication under the following conditions: 80 watts for 10 s, followed by a 15 s rest, repeated 10 times. Post-ultrasonication, the mixture was boiled for 5 min. After cooling to room temperature, it was centrifuged at 14,000× *g* for 10 min at 4 °C. SDS-PAGE analysis was conducted with 5% stacking and 10% separating gels, where the supernatant from the first day and 10 μL of boar blood serum were loaded. Additional SDS-PAGE analyses were performed for egg supernatants collected on days 0, 7, 14, and 21 using gels of the same proportions.

### 2.4. Protein Bands Cutting and Protein Re-Collection

To investigate protein changes during embryonic development, prominent protein bands from day-zero eggs were excised and analyzed using the Bradford Protein Assay Kit (Beyotime Biotechnology, Shanghai, China). These samples underwent qualitative analysis via LC-MS/MS following a standardized protocol.

### 2.5. LC-MS/MS for First-Day Eggs and Re-Collected Proteins from Bands

Protein digestion was performed for eggs at different incubation days and for proteins collected on the first day using the FASP method [30]. For each sample, 200 μg of protein underwent ultrafiltration (Pall, 10 kD) to remove detergents, DTT, and other low-molecular-weight contaminants. The samples were repeatedly centrifuged with 200 μL of UA buffer (8 M urea, 150 mM Tris-HCl, pH 8.0). Following impurity removal, 100 μL of 0.05 M iodoacetamide in UA buffer was added to block reduced cysteine residues, and samples were incubated in the dark for 20 min. The filters were washed three times with 100 μL of UA buffer and twice with 100 μL of 25 mM NH4HCO3. Proteins were then digested overnight at 37 °C with 3 μg of trypsin (Promega, Madison, WI, USA) in 40 μL of 25 mM NH4HCO3, and the resulting peptides were collected.

### 2.6. Chromatographic Fractionation

Chromatographic separation was carried out using a column (0.15 mm × 150 mm, RP-C18, Column Technology Inc., Lombard, IL, USA) equilibrated with 95% buffer A (0.1% formic acid in water). Peptides were loaded onto Zorbax 300SB-C18 peptide traps (Agilent Technologies, Wilmington, DE, USA) and eluted with buffer B (0.1% formic acid, 84% acetonitrile [ACN]) using a linear gradient: 4% to 50% buffer B over 50 min, followed by 50% to 100% buffer B over 5 min, and held at 100% for 6 min.

The peptide eluates were analyzed using a Q Exactive mass spectrometer. The survey scan ranged from *m*/*z* 300 to 1800, capturing positive ions. For MS1, the resolution was set at 70,000 (at *m*/*z* 200), the AGC target was 3 × 10^6^, the maximum injection time was 10 ms, and the dynamic exclusion was 40 s. Each scan selected 10 fragment ions for MS/MS, using higher-energy collisional dissociation (HCD) with a normalized collision energy of 30 eV and an under-fill ratio of 0.1%. The entire experiment was conducted in triplicate to ensure reproducibility.

### 2.7. PRM-Dynamic Changes in Forty Egg Proteins During Egg Development

The stable isotope iRT KIT peptide (Biognosys AG, Zurich, Switzerland) was added to each digested peptide sample across different incubation times, with three replicates per condition, serving as internal standards. Samples (2 μg in 40 μL of 0.1% formic acid buffer) underwent desalination using stage-tip-mounted C18 cartridges (Sigma-Aldrich, St. Louis, MO, USA) before reversed-phase chromatography on an nLC-1200 system (Thermo Fisher Scientific, Waltham, MA, USA). Liquid chromatography (LC) was performed with gradients of 5% to 35% ACN over 45 min. PRM analysis was conducted on a Q Exactive Plus mass spectrometer (Thermo Fisher Scientific, Waltham, MA, USA).

The full scan was performed with a 60 min analysis time in positive ion mode, covering a scan range of 300–1800 *m*/*z*, at a resolution of 60,000 (@*m*/*z* 200), with an AGC target of 3 × 10^6^ and a maximum injection time of 200 ms. This was followed by a PRM event using an isolation window of 1.6 Th, a resolution of 30,000 (@*m*/*z* 200), an AGC target of 3 × 10^6^, and a maximum injection time of 120 ms. HCD was employed for MS2 activation, with a normalized collision energy set at 27 eV. Raw data analysis was conducted using Skyline (MacCoss Lab, University of Washington, Seattle, WA, USA), with peptide signal intensity quantified based on the peak area for each sample and normalized to internal standards.

### 2.8. MS Data Analysis

Raw data files were searched against the *H. hystricis* transcriptome using Mascot (version 2.2), employing trypsin as the enzyme with up to two allowed missed cleavages. Carbamidomethylation was set as a fixed modification for cysteines, while N-terminal acetylation and methionine oxidation were treated as variable modifications. The decoy database was set to reverse, with peptide and fragment mass tolerances set at ±20 ppm and 0.1 Da, respectively, and a Mascot score threshold of ≥20. A global false discovery rate of 0.01 was applied for peptide and protein identification. Identified polypeptides/proteins were further queried against a custom *H. hystricis* protein database compiled from transcriptomic data (NCBI: PRJNA1168713), and homologous protein searches were conducted using the UniProt database (https://www.uniprot.org/, accessed on 27 November 2024).

Label-free quantification was carried out using MaxQuant (version 1.6.14.0, https://maxquant.net/maxquant/, accessed on 27 November 2024) according to Tyanova et al. [31], with protein abundance assessed by intensity-based absolute quantification (iBAQ). Protein levels were determined based on normalized spectral intensities.

## 3. Results and Discussion

Ticks pose a considerable threat to a wide range of terrestrial vertebrates, including mammals, birds, reptiles, and amphibians. As obligate hematophagous parasites, they cause adverse effects such as anemia, malnutrition, and damage to the host’s integument. They are also vectors for various pathogens, including fungi, viruses, and bacteria (e.g., rickettsiae), leading to diseases like Lyme disease, Rocky Mountain spotted fever, and Mediterranean spotted fever. Given the need for sustainable tick control strategies, vaccination-based approaches are recognized as more environmentally friendly compared to chemical methods. Since tick eggs contain all the essential components for embryonic development, they may act as reservoirs for protective antigens, making the targeting of proteins during the egg incubation phase a promising strategy for tick control.

### 3.1. Protein Expression Profile at Different Egg Developmental Stages

As shown in Figure 1, SDS-PAGE analysis revealed that tick egg proteins ranged in size from 15 to 250 kDa, with bands 3, 4, 5, and 6 being the most prominent. LC-MS/MS analysis of day-one egg samples identified vitellogenins (Vgs) as the predominant proteins in these high-intensity bands (Figure 2). Protein band intensity remained relatively unchanged across different incubation days for bands 3, 4, 5, and 6, while gradual reduction was observed for bands 1, 2, 8, 10, and 11. Notably, bands 7 and 12 showed reduced intensity by the 14th day of incubation, whereas band 9 exhibited increased intensity at day 7.

To identify the proteins in these bands, all 12 bands were excised, and the proteins were extracted using a Protein Extraction Kit (Bio-Rad, Hercules, CA, USA). The re-collected proteins were analyzed via LC-MS/MS, and after annotation, they were ranked according to their iBAQ values. The results revealed that 9 out of the 12 bands predominantly contained vitellogenin (Cluster-17602.25911) and vitellin-a (Cluster-17602.43418), indicating that vitellogenin serves as the primary nutrient source in day-one eggs. Furthermore, these results suggest that the ovary may directly absorb exogenous vitellogenin from the adult female tick. This is supported by the work of Yang et al., who demonstrated that a process of protein auto-synthesis occurs during the early stages of ovarian development, with developing oocytes absorbing vitellogenin from the hemolymph of the adult female tick.

Other prominent proteins identified included actin 1 (Cluster-17602.36441), Kunitz domain-containing protein 1 (Cluster-17602.30986), yolk protein (Cluster-17602.15229), and cathepsin D (Cluster-17602.33592). Additionally, minor proteins such as histone H4 (Cluster-17602.34926), cysteine-rich protein (Cluster-17602.29432), and serpin (Cluster-11893.0) were also detected (Figure 2). These proteins are known to participate in processes like anti-inflammation, anti-hemostasis, and immune responses in adult ticks, though their roles in tick eggs remain to be clarified.

### 3.2. Identification of High-Confidence Polypeptides and Their Dynamics Changes in Egg Development

The LC-MS/MS analysis of day-one tick egg samples searched against a protein library derived from the *H. hystricis* transcriptome and identified 1416 unique peptides and 207 polypeptides. Among these, 125 sequences were classified as high-confidence transcripts, with each unique peptide being matched at least twice. BLAST annotation against the tick database in UniProt revealed 108 transcripts, of which 43 sequences had over 80% identity with known proteins, 28 exhibited 60–80% identity, and 37 showed lower identity with previously released proteins.

The abundance of these high-confidence proteins, based on iBAQ, varied significantly. Notably, 16 proteins had iBAQ values exceeding 1.00E+10, indicating their high abundance in day-one egg samples (Table 1).

To further investigate the roles of these proteins during embryonic development, Parallel Reaction Monitoring (PRM), a highly sensitive and reproducible mass spectrometry quantification technique, was employed to measure the levels of 40 target proteins across four stages of egg incubation (0, 7, 14, and 21 days) (Appendix A). This approach allowed for precise monitoring of protein abundance changes throughout tick oogenesis.

The 40 selected proteins included 18 enzymes, five serpins, one Kunitz domain-containing protein, one α_2_-macroglobulin, two neutrophil elastase inhibitors (NEI), three heat shock protein 70 (HSP70), four vitellogenins (Vgs), three cytoskeletal proteins, one ferritin, one elongation factor, and one uncharacterized protein (Figure 3). These targets were chosen based on their potential importance in egg development and to observe their dynamic expression during the different stages of tick embryogenesis. 

### 3.3. Classification and Potential Function of Egg Proteins

Based on their functions and families, the 108 transcripts identified were classified into eight categories: transporters (20), enzymes (28), immunity and antimicrobial-related proteins (7), proteinase inhibitors (20), cytoskeletal proteins (5), heat shock proteins (3), secreted proteins (9), and uncharacterized proteins (16) (Appendix A). These categories play essential roles in processes critical to tick egg development, including nutrient supply, metabolism, molecular synthesis and transfer, as well as antimicrobial defense and immunity.

Among the 20 enzymes identified were cysteine proteases, aspartic proteinases, hydrolases, dehydrogenases, and peroxidases. LC-MS/MS analysis of day-one eggs showed that only yolk cathepsin and heme-binding aspartic proteinase (Cluster-11877.0) had high abundances, with iBAQ values exceeding 1.00E+10. The remaining enzymes displayed generally lower iBAQ values. Given the substantial diversity of egg proteins and the complex changes occurring throughout egg development, these enzymes are discussed in separate groups to provide a more detailed analysis.

### 3.4. Vgs

Vg, a yolk protein precursor synthesized in the fat body of female ticks, is secreted into the hemolymph and absorbed by developing oocytes. Vitellin (Vt), derived from Vg, acts as the primary nutrient and energy source during embryogenesis, playing a pivotal role in tick egg development.

The analysis identified one Vt and five Vg proteins as major components in day-one eggs, with Vt (Cluster-17602.43418) and one Vg (Cluster-17602.25911) showing particularly high abundances (iBAQ > 1.00E+10), suggesting a dependence on Vg and Vt throughout egg development (Figure 2 and Figure 3A).

The presence of Vg and Vt across nine distinct bands, corresponding to molecular weights of approximately 16, 40, 45, 50, 63, 70, 110, 200, and 280 kDa, indicates that Vg undergoes digestion into various fragments. This observation supports the findings by Silveira et al. [32], who described Vg as a hemolymphatic phospholipoglycoprotein. Inside the oocyte, Vg is cleaved to produce Vt, consisting of nine polypeptides with molecular weights of 203, 147, 126, 82, 74, 70, 61, 47, and 31 kDa [33]. Thus, Vt is likely generated from Vg during the incubation period. This dynamic relationship is further supported by LC-MS/MS data (Figure 2), showing that the band intensities of Vg and Vt remain relatively stable during development, as seen in bands 3, 4, 5, and 6 in Figure 1. The consistent presence suggests that Vg degradation is an ongoing process, with continuous conversion to Vt throughout incubation. For instance, band 9 showed increased intensity on day 7, indicating that one of the increasing proteins may be Vt, initially detected on day 0, accounting for only 13% of the total protein.

Considering Vg’s critical role in egg development, PRM was employed to examine its concentration variations across different developmental stages. The PRM results showed that Vg levels fluctuated 1-4-fold over the four incubation stages (Figure 3A), suggesting that Vg is increasingly digested into Vt to sustain the demands of egg development.

### 3.5. Lipocalins

Lipocalins are low-molecular-weight proteins primarily responsible for transporting small molecules [29]. In soft ticks, the lipocalin family is notably abundant in terms of both the number of members and expression levels [34,35,36,37,38]. This study identified 13 lipocalin-annotated transcripts in day-one tick eggs, with sequence identities ranging from 40% to 70% compared to other tick species, indicating considerable sequence variability among lipocalins [39].

While lipocalins are known to interact with histamine, serotonin, and leukotriene B4 to reduce host inflammation in ticks [40,41,42] and are predominantly expressed in salivary glands and secreted into saliva [14,35,36], their role in tick eggs appears different. The identified lipocalins were present at low levels in day-one eggs, suggesting that their function may involve the transport of nutritional small molecules or other processes supporting embryonic differentiation and growth [40] rather than the primary anti-inflammatory role observed in adult tick saliva [43,44,45,46,47].

Despite limited knowledge about lipocalin function in tick eggs, one study on recombinant lipocalin (rHlLIP) from *Haemaphysalis longicornis* has shown its ability to reduce tick hatch rates, indicating a potentially critical role in tick development [48]. Therefore, further research is necessary to elucidate the specific functions of lipocalins during tick embryogenesis.

### 3.6. Cysteine Proteases

Proteases are vital for protein hydrolysis in organisms and are categorized based on their catalytic mechanisms into serine, cysteine, aspartic, threonine, and metalloproteases. This study identified four cysteine proteases and six aspartic proteases in day-one tick eggs, both of which play key roles in protein hydrolysis, particularly in the degradation of vitellin during embryonic development.

In parasitic organisms, cysteine proteases belong to the papain family and are divided into two major groups: clan CA [49] and clan CD [50]. These enzymes are involved in various biological processes, such as protein degradation and digestion [51].

Among the identified cysteine protease sequences, three showed 78%, 82.2%, and 92.4% identity with known tick midgut cysteine proteases, though their abundance in day-one eggs was low. Given the substantial amount of vitellin or yolk protein needed for egg development, a single cysteine protease is insufficient for complete digestion, suggesting that additional proteases from this family or related families contribute to nutrient processing. For instance, cathepsin L, a cysteine protease known for its potent haemoglobinase activity [52,53,54], plays a critical role in the hydrolysis of vitellin [20,55,56]. In this study, cathepsin L (Cluster-17602.36868) remained relatively stable during the first three stages of egg development, but its quantity increased significantly by 1.65-fold on day 21, as indicated by PRM results (Figure 3J). Notably, cathepsin was the main component of Band 7, constituting 81% of its composition (Figure 2). By day 14, Band 7 showed a marked reduction in protein abundance (Figure 1), suggesting that cathepsin levels decrease in later stages, potentially due to reduced demand for cathepsin activity as other proteases take over roles in preparation for larval hatching. The identification of a diverse array of proteases in tick eggs indicates the presence of a complex digestive cascade that facilitates nutrient digestion during embryogenesis.

### 3.7. Aspartic Proteases

Similar to cysteine proteases, the aspartic proteinase family is also essential for protein digestion and represents a major enzymatic component in the tick gut [57]. The breakdown of hemoglobin in ticks relies on a network of cysteine and aspartic proteinases [52]. In this study, four aspartic proteinases and two cathepsin D proteins were identified, both belonging to the aspartic proteinase family. Like cysteine proteases, aspartic proteinases participate in vitellin hydrolysis, showing peak activity at a heme-to-globin ratio of 1:1, which suggests that vitellin degradation may be regulated based on heme availability [19].

Two heme-associated aspartic proteinases were detected, with PRM results indicating an increase in one heme-binding aspartic proteinase on day 21 of incubation, while the other remained stable throughout egg development. These variations in expression likely contribute to the diversity of hydrolytic activities within the aspartic proteinase family. Previous research has shown that aspartic proteinase precursor (BYC) exhibits a slow rate of vitellin hydrolysis [58] due to the absence of a critical aspartic acid residue, in contrast to the egg cysteine endopeptidase (VTDCE), which demonstrates higher hydrolytic activity toward vitellin [59].

Cathepsin D, a key lysosomal aspartic protease, exhibited a slight decline in abundance during egg development according to PRM analysis, with both identified isoforms (Cluster-18960.0 and Cluster-17602.1417) showing reduced levels as development progressed (Figure 3B). This finding is consistent with findings by [60], which noted a gradual decrease in cathepsin D activity after initial egg incubation. This trend contrasts with bisexual *H. longicornis*, where cathepsin D expression and activity peak on days 11 and 13, respectively [61].

The fluctuating levels of cathepsin expression and activity across tick species suggest that cathepsin D may have a stage-specific regulatory role in egg development. In our study, cathepsin D (Cluster-17602.33592) was a major component of bands 8 (69%) and 9 (8%), with band 8 showing a slight decline in protein abundance after day 14 of incubation (Figure 2). This decline may indicate a reduced demand for cathepsins in the later stages of egg development, potentially compensated by other cysteine and aspartic proteases, reinforcing the hypothesis that a complex digestive cascade is present in tick eggs.

### 3.8. GSTs

During blood feeding, ticks are exposed to elevated levels of reactive oxygen species (ROS) due to the host blood’s high iron content. As ticks concentrate the ingested blood, iron levels increase, which can react with oxygen in the tick’s body, leading to the formation of ROS, such as hydrogen peroxide (H_2_O_2_) [62]. H_2_O_2_ poses a significant threat to aerobic organisms by causing extensive damage to membrane lipids, nucleic acids, and proteins [63].

To mitigate the harmful effects of ROS and ensure survival, ticks employ both enzymatic and non-enzymatic mechanisms to reduce oxidative damage. Key detoxification enzymes, such as catalase, glutathione S-transferases (GSTs), and glutathione peroxidase, catalyze the conversion of ROS into less harmful molecules.

In this study, four sequences related to enzymatic detoxification were identified, including one (Cluster-17602.35954) showing 78% identity with GST. GSTs play a critical role in managing chemical toxicity and environmental stress, particularly in response to insecticide exposure. Although LC-MS/MS results indicated that GST was not a predominant protein on the first day of egg incubation, PRM analysis showed a marked increase in its expression during the late incubation stages, with a 4.12-fold rise by day 21 (Figure 3C). This pattern aligns with findings in *Boophilus microplus* [22], where GST levels increased progressively during embryonic development, peaking on day 20, which corresponded with the highest GST activity observed during this period. Knockdown experiments have shown that reduced GST expression increases susceptibility to insecticides in both larvae and adult male ticks, leading to higher mortality rates, decreased egg-laying capacity, and lower egg weight [64,65]. These results indicate that elevated GST levels are essential for successful larval hatching.

Beyond detoxification, GSTs are also involved in transporting endogenous hydrophobic compounds, such as hormones, steroids, and hemoglobin [66], and they play a role in viral defense mechanisms [67]. This multifunctionality underscores the significance of GSTs in tick eggs, suggesting they contribute to various physiological processes essential for embryonic development and survival.

### 3.9. Catalase

Unlike GSTs, which are involved in detoxification, catalase plays a key role in reducing oxidative stress in ticks by breaking down H_2_O_2_ [22]. H_2_O_2_ is generated in the tick midgut, where high concentrations of hemoglobin, iron, and oxygen are present. In tick eggs, two catalase-related sequences, Cluster-17602.16856 and Cluster-17602.16862, have been identified, exhibiting 73.2% and 73.9% identity with catalase, respectively. The abundance of these sequences increased during egg development, with Cluster-17602.16862 showing a marked surge on day 21 (Figure 3C), indicating elevated H_2_O_2_ levels and rising oxidative stress throughout the developmental stages.

To mitigate the toxicity of H_2_O_2_, ticks upregulate antioxidant production to combat oxidative damage, thereby ensuring proper egg development and hatching. This mechanism is consistent with observations in other organisms, where catalase catalyzes the conversion of H_2_O_2_ into water and oxygen [68]. The essential role of catalase in egg development is further supported by Kumar et al. [69], who found that silencing the catalase gene resulted in decreased egg weight and reduced hatching rates.

Catalase is also implicated in transovarial virus transmission. Its gradual increase during egg development may help limit viral proliferation. Budachetri et al. [70] reported a 2- to 11-fold rise in catalase expression upon viral infection and observed a 44% reduction in transovarial virus transmission following catalase knockdown

### 3.10. Peroxiredoxins(Prx)

Prx also functions as a H_2_O_2_ scavenger [71]. Studies demonstrate that Prx, which possesses conserved cysteine residues, efficiently neutralizes H_2_O_2_ [72]. One identified sequence showed 96.6% identity with *H. longicornis* Prx, confirming the annotation. In *H. longicornis*, Prx (Cluster-17602.35118) exhibited a 2-fold increase just before larval hatching (Figure 3C), likely due to elevated lipid peroxidation and rising oxygen consumption during late egg development Freitas et al. [22]. The upregulation of Prx is essential for successful egg development and hatching [72]. Prx knockdown experiments revealed elevated H_2_O_2_ levels in *Ixodes scapularis* cell lines treated with paraquat, highlighting its vital role in managing oxidative stress [73].

Similar to catalase, Prx is associated with virus transmission, although one form of Prx has been suggested to facilitate viral replication through mechanisms beyond H_2_O_2_ clearance [74]. These findings emphasize the multifunctional roles of antioxidant enzymes in tick eggs, underscoring their importance in oxidative stress management and potentially influencing viral dynamics.

### 3.11. Dehydrogenase

Glyceraldehyde-3-phosphate dehydrogenase (GAPDH) is a critical enzyme in the glycolytic pathway, the primary metabolic route for energy production from glucose.

One GAPDH was identified in day-one tick eggs, with levels steadily increasing throughout incubation, indicating a growing need for GAPDH to meet the energy demands of egg development. This observation aligns with findings by Hildebrandt et al. [75], which reported that GAPDH catalyzes the conversion of glyceraldehyde-3-phosphate to 1,3-bisphosphoglycerate, thereby facilitating ATP production.

Additionally, aldehyde dehydrogenase (Cluster-17602.38580), another relevant enzyme, showed an 11-fold increase in expression by day 21 of egg development (Figure 3D). Although its role in ticks remains poorly understood, the significant upregulation suggests it plays a pivotal role in supporting egg development.

### 3.12. Hydrolases

Nine hydrolases were detected in day-one eggs, with five selected for further investigation using PRM to monitor dynamic changes during development. An upward trend was observed in three hydrolases: neurotrypsin (Cluster-17602.21941), ATP synthase subunit beta (Cluster-17602.34511), and alpha-mannosidase (Cluster-17602.36136).

Neurotrypsin levels surged 15-fold by day 7 compared to day-one levels, while ATP synthase subunit beta increased on day 7 and peaked on day 21 during the larval stage. Alpha-mannosidase exhibited a 2-fold rise by day 7, maintaining this level through day 21 (Figure 3E). In contrast, carboxypeptidase and phospholipase showed no significant variation from day-one levels. The upregulation of most hydrolases during egg development indicates an elevated requirement for nutrient digestion, reflecting the increasing metabolic demands associated with embryogenesis and preparation for hatching

### 3.13. AMP

Innate immunity is a fundamental aspect of tick biology that significantly impacts development. It is regulated by various proteins, particularly antimicrobial peptides (AMPs) and serpins.

AMPs are essential effector molecules in the tick immune system, capable of neutralizing pathogenic microorganisms and playing a key role in innate immune defense [76]. Numerous tick AMPs, such as microplusin, VTDCE, defDM, Hb98-114, and defensin, have been detected in tissues, including the salivary glands, hemolymph, and midgut [77,78,79,80].

In this study, several AMPs with antimicrobial activity were identified, including microplusin, alpha-2-macroglobulin protein (α_2_M), cysteine-rich protein, peroxinectin, and histidine-rich proteins. Cluster-17602.36041 showed an 80% similarity to α_2_M from Amblyomma cajennense, and proteomic analysis revealed an 11-fold and 15-fold increase in its levels at days 7 and 14, respectively, with a significant rise by day 21 compared to day-one eggs (Figure 3J). This upregulation suggests enhanced innate immune responses during late egg development, as α_2_M protects against harmful proteases from various sources, including pathogens [81]. Buresova et al. [82] demonstrated that silencing α_2_M in *Ixodes ricinus* through dsRNA interference reduced the phagocytic activity of hemocytes against *Chryseobacterium indologenes*, both in vitro and in vivo.

Cluster-17602.29432 exhibited high similarity to cysteine-rich proteins and a notably high iBAQ value of 1.11E+09 in day-one eggs, suggesting a significant role in innate immunity during early egg development. Cysteine-rich proteins are known to inhibit Gram-negative bacteria and proteolytic enzymes such as chymotrypsin and elastase [83,84].

In contrast, Clusters 17602.7590 and 17602.7506, which showed lower identity with microplusin and peroxinectin, also had lower iBAQ values. Microplusin, a cysteine-rich AMP, exerts its antibacterial effects through metal binding [85] but was present in lower quantities in day-one eggs. Similarly, peroxinectin levels were low in day-one eggs and remained underexplored beyond its initial identification in *Rhipicephalus microplus* [86]. These two proteins have not been analyzed through PRM trials, leaving their expression dynamics during egg development unknown. Considering the involvement of multiple proteins in tick egg innate immunity, further research is necessary to investigate the mechanisms and functional roles of these immunity-related proteins in this context. Additionally, two transcripts showed 38% and 40% identity with his-rich 1 fat body overexpressed and were categorized as immunity-related proteins. This classification is based on the presence of histidine-rich residues in many AMPs at their C- and N-termini [85,87]. Given the complexity of immunity-related proteins in tick eggs, further research is needed to elucidate the mechanisms underlying their roles in tick innate immunity during development.

### 3.14. Serpin

Proteinase inhibitors are categorized into at least 18 families based on their primary and tertiary structures and inhibition mechanisms [88]. In ticks, four main groups of serine protease inhibitors have been identified: Kunitz domain inhibitors, Kazal domain inhibitors, trypsin inhibitor-like cysteine-rich domain (TIL) inhibitors, and serpins [89].

Ticks are particularly rich in proteinase inhibitors, especially serpins. This study identified 20 sequences annotated as proteinase inhibitors in tick eggs, including eight serpins, five Kunitz domain-containing proteins, and six NEIs.

Seven of the sequences were classified as serpins, showing 54% to 94% identity. Two serpins, Cluster-17602.31952 and Cluster-11893, were highly abundant in day-one eggs, with iBAQ values of 1.18E+09 and 4.20E+09, respectively. LC-MS/MS analysis confirmed that these serpins were predominant in bands 7, 11, and 12, accounting for 5%, 5%, and 10% of the total protein content in these bands (Figure 2). While protein abundance in band 11 remained stable throughout the four stages of egg development, bands 7 and 12 showed a rapid decline in protein levels by day 14. This pattern suggests the primary proteins in these bands are being utilized, whereas the serpins are not, as evidenced by a three-fold increase in Cluster-17602.31952 abundance by day 21, according to PRM results (Figure 3F).

The high abundance and upregulation of these serpins during tick egg development imply a significant role in reproductive processes. A similar trend was reported in *Rhipicephalus haemaphysaloides*, where serpin RHS8 mRNA expression increased from the feeding stage to oviposition. RNA interference (RNAi) experiments demonstrated that serpin silencing led to reduced body weight, extended feeding duration, fewer eggs laid, and lower egg hatchability [90,91], underscoring the critical role of serpins in tick reproduction.

Serpins are well-established anti-inflammatory proteins in adult ticks, known for their ability to modulate the host’s immune response by influencing immune cell migration to sites of inflammation [92]. However, tick eggs do not encounter the host’s immune system. It is hypothesized that components of the host’s blood transferred to the eggs may not be completely metabolized. In tick eggs, serpins might serve a similar function as in adults, providing anti-inflammatory effects that support development and hatching. This mechanism likely involves interactions with proteases such as serine protease, cathepsin G, thrombin, and chymotrypsin. For example, RmS-15 inhibits thrombin activity [93], while *Amblyomma americanum* serpin AAS19 inhibits both cathepsin G and thrombin [94].

Additionally, serpins act as anticoagulants by suppressing coagulation factors [94] or by blocking the intrinsic and common pathways of the coagulation cascade [95]. Noteworthy examples include serpins such as Iris [96], Iripin-8 [95], and AAS19 [97], which inhibit coagulation factors like FXa, FVIIa, and FIXa.

As egg development progresses, particularly during the late stages when larvae are about to hatch, the demand for serpins may increase to prevent hemolymph coagulation in the emerging larvae. Proper regulation of coagulation is critical for successful hatching and survival, as it can prevent blockages that could hinder larval development. Understanding the specific functions of serpins during this critical phase could provide valuable insights into tick physiology and adaptive strategies in response to environmental pressures.

Serpins are also associated with anti-complement activities [98,99,100,101] and may facilitate viral colonization [102]. However, whether serpins play a role in preventing the transovarial transmission of pathogens during egg development remains uncertain.

### 3.15. Kunitz Domain-Containing Protein

The Kunitz domain-containing protein, known for its thrombin-inhibiting activity, is part of the serpin family. In this study, five Kunitz domain-containing proteins exhibited low sequence identity (35–46%) compared to those from *Rhipicephalus microplus*. PRM results indicated that one Kunitz protein (Cluster-17602.19391) was downregulated during the egg protein stage (Figure 3F). SDS-PAGE and LC-MS/MS analyses identified another Kunitz protein (Cluster-17602.30986) as prominent in bands 10, 11, and 12, constituting 61%, 64%, and 9% of these bands, respectively (Figure 2). As egg development progressed, the abundance of these bands decreased, suggesting digestion of the primary Kunitz domain-containing proteins. The potential function of Kunitz proteins during egg development remains unclear in the current study, despite the extensive study of other Kunitz proteins for anticoagulant properties, such as BmTI from *Boophilus microplus*, Doenitin-1 from *Haemaphysalis doenitzi* [103], Boophilin from *Rhipicephalus microplus* [104,105], and Hemalin from *H. longicornis* [106]. Kunitz proteins may be involved in anticoagulation in tick eggs, but the reason for the decrease in Kunitz abundance remains unclear. It is possible that different Kunitz family members take on anticoagulant roles at various stages of egg development. In this context, further investigation into the functional roles of the Kunitz protein is warranted.

### 3.16. Neutrophil Elastase Inhibitors (NEI)

More than half of the immune-related proteins identified in this study were NEIs, with iBAQ values indicating high abundance: three exceeded 1.00E+10, two ranged between 100 and 1000 (including Cluster-17602.36134 and Cluster-17602.34822), and one was below 100 (Cluster-17602.33042). Notably, the expression levels of two NEIs (Cluster-17602.36134 and Cluster-17602.33042) increased by 1.3- to 1.8-fold in day-21 eggs, suggesting their involvement in egg development (Figure 3G).

The high abundance of NEIs in tick eggs implies a significant role in immunity. NEIs, which are enzymes produced by host immune cells, degrade proteins as part of the defense against pathogens and may inhibit neutrophil elastase activity [107], potentially enhancing defenses against parasitic threats. The immune function of NEIs has also been observed in the eggs of the blood-feeding insect Triatoma infestans [108].

NEIs belong to the pacifastin family, which is part of the conserved serpin superfamily in arthropods [108,109], indicating functional similarities with other serpins. However, their specific roles in ticks are still largely unexplored. Further research is required to fully elucidate the potential contributions of NEIs to tick development and immune regulation.

### 3.17. HSP

The heat shock protein (HSP) family plays a critical role in helping cells and organisms cope with high temperatures and various stress conditions, including exposure to toxins and pathogen infections. Numerous heat shock proteins have been reported in insects [110,111,112,113,114]. In ticks, HSP70, a prominent member of this family, has been linked to pathogen infections and exhibits anticoagulant activity [115,116,117]. Our findings indicate that HSP70 expression is upregulated during tick egg development (Figure 3H), suggesting that increased levels are necessary as hatching approaches. However, the specific functions of HSP70 in tick eggs remain poorly understood.

### 3.18. Cytoskeletal Protein and Other Proteins

Among cytoskeletal proteins, a significant upregulation (4-6-fold) was observed by day 14, implying an increased demand for these proteins to support larval growth, particularly as tissue formation begins around day 7 [118] (Figure 3I). Within the groups of secreted and uncharacterized proteins, three were found to be highly abundant in day-one eggs, though their precise identities remain undetermined. PRM analysis of 40 proteins revealed that one uncharacterized protein (Cluster-17602.39905) showed a dramatic 43-fold increase by day 21 compared to day one (Figure 3J). Additionally, Elongation Factor 1-alpha (Cluster-17602.36724) exhibited the most significant rise, with a 292-fold increase by day 21 (Figure 3J). Despite limited insights into their functions in ticks, the substantial abundance of these proteins suggests they play critical roles in egg development, warranting further functional studies.

## 4. Conclusions

This study identified a broad range of proteins in tick eggs, with transporters, enzymes, and proteinase inhibitors accounting for half of the total. These proteins are likely involved in nutrient supply, detoxification, protein digestion, and innate immunity. The extensive diversity and dynamic changes in the proteome of *H. hystricis* eggs, from day one of incubation to day 21, may aid in identifying new targets for vaccine development, for example, catalase, cysteine protease and aspartic protease, AMPs, etc. They are important for egg development. Therefore, tracking their dynamic changes during embryonic development can be a key research direction for further studies, such as RNA interference and protein expression in vitro for anti-tick vaccine. By developing vaccines targeting these proteins, it may be possible to interfere with egg development, thereby suppressing tick population growth and reducing their ability to transmit pathogens. Though these proteins are not directly exposed to the environment during the tick lifecycle, they remain promising vaccine candidates.

The specific roles of some egg proteins remain unclear, such as Kunitz domain-containing proteins, dehydrogenases, and hydrolases, but this research offers a comprehensive overview of egg proteins and lays the foundation for future studies.

## Figures and Tables

**Figure 1 animals-14-03466-f001:**
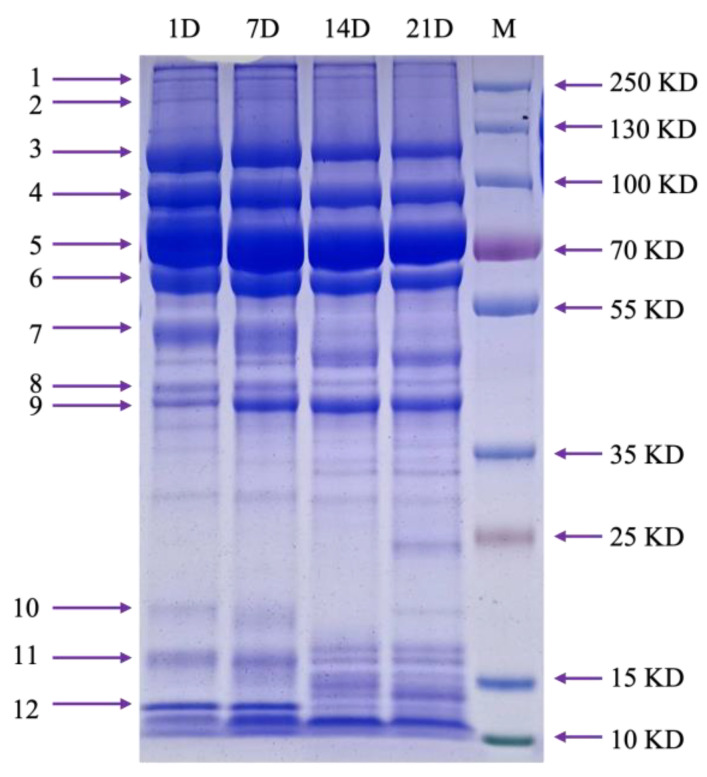
SDS-PAGE analysis of protein extract from *H. hystricis* eggs. A total of 80 μg of protein extract was loaded per sample. Lane M represents the 15–250 kDa molecular weight marker. D1, D7, D14, and D21 correspond to eggs incubated for 1, 7, 14, and 21 days, respectively. Bands 1–12 indicate the protein bands excised for further analysis.

**Figure 2 animals-14-03466-f002:**
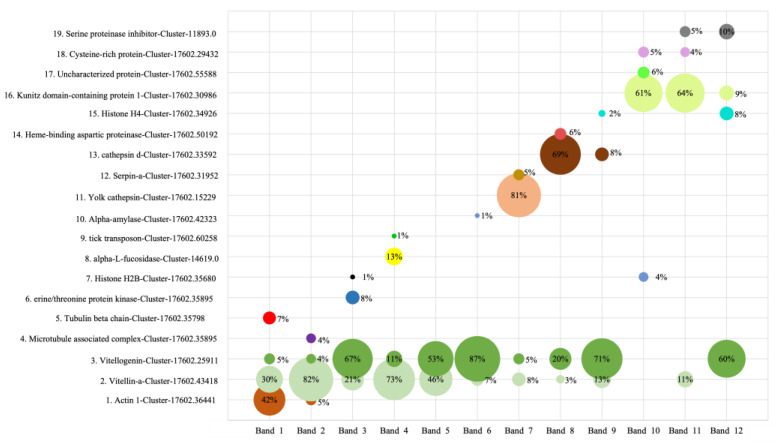
Top 4 most prominent proteins identified in 12 bands from day-one eggs.

**Figure 3 animals-14-03466-f003:**
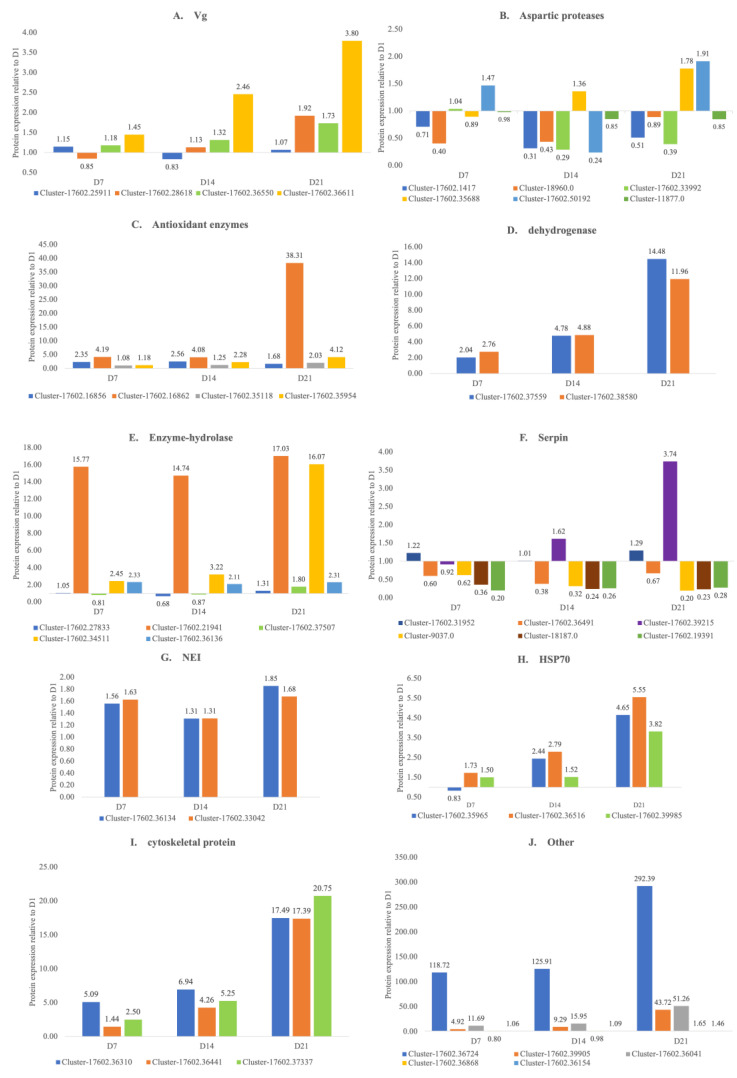
Analysis of protein dynamics in *H. hystricis* tick eggs across different incubation days. Each sample was spiked with the stable isotope iRT KIT peptide as an internal standard. Tryptic peptides were analyzed using the nLC-1200 system. Protein abundances at 7, 14, and 21 days of incubation were normalized to the levels observed at day. D1, D7, D14, and D21 correspond to eggs incubated for 1, 7, 14, and 21 days, respectively.

**Table 1 animals-14-03466-t001:** High abundance transtripts (iBAQ values exceeding 1.00 × 10^10^) identified in eggs of *H. hystricis* on day-one incubation.

No	Transcripts ID in PRJNA1168713	Alignment Entry and Overview	E Value	Score	Identity (%)	iBAQ
TRANSPORTERS	Cluster-17602.43418	A0A8X8MG03, Vitellin-a, *Haemaphysalis flava*	0	9.17 × 10^3^	93.30%	1.51 × 10^10^
	Cluster-17602.25911	A0A411G179, Vitellogenin, *Haemaphysalis flava*	0	8.9 × 10^3^	87.50%	1.39 × 10^10^
ENZYMES	Cluster-17602.15229	A0A6M2CKA2, Yolk cathepsin, *Rhipicephalus microplus*	0	1.37 × 10^3^	66.80%	7.67 × 10^9^
	Cluster-11877.0	A0A8K1PH81, Heme-binding asparlic proteinase, *Rhipicephalus microplus*	0	1.35 × 10^3^	67.50%	1.40 × 10^9^
IMMUNITY-RELATED PROTEIN	Cluster-17602.29432	A0A5B9BYB0, Cysteine-rich protein, *Haemaphysalis flava*	3.10 × 10^110^	8.10 × 10^2^	86.70%	1.11 × 10^9^
	Cluster-17602.8619	A0A6M2D6P5, His-rich 1 fat body overexpressed, *Rhipicephalus microplus*	5.10 × 10^17^	1.88 × 10^2^	38.10%	1.50 × 10^9^
	Cluster-17602.7590	A0A6M2D6P5, His-rich 1 fat body overexpressed, *Rhipicephalus microplus*	1.80 × 10^17^	1.91 × 10^2^	40.20%	2.16 × 10^9^
PROTEANASE INHIBITORS	Cluster-17602.31952	A0A5P8H6S1, Serpin-a, *Haemaphysalis longicornis*	7.40 × 10^158^	1.17 × 10^3^	61.20%	1.18 × 10^9^
	Cluster-11893.0	A0A034WTW4, Serine proteinase inhibitor, *Rhipicephalus microplus*	1.10 × 10^29^	2.69 × 10^2^	56.80%	4.20 × 10^9^
	Cluster-17602.30986	A0A034WTW0, Kunitz domain-containing protein 1, *Rhipicephalus microplus*	0	1.83 × 10^3^	34.50%	4.49 × 10^9^
	Cluster-17602.10962	A0A8F1NJE0, Neutrophil elastase inhibitor, *Haemaphysalis flava*	5.70 × 10^30^	2.70 × 10^2^	53.50%	1.63 × 10^9^
	Cluster-17602.18555	A0A8F1NJE0, Neutrophil elastase inhibitor, *Haemaphysalis flava*	7.40 × 10^49^	3.94 × 10^2^	73.50%	1.25 × 10^10^
	Cluster-17602.35730	A0A8F1NJE0, Neutrophil elastase inhibitor, *Haemaphysalis flava*	6.40 × 10^47^	3.92 × 10^2^	72.50%	1.15 × 10^10^
OTHERS	Cluster-17805.0	B7QGW1, Secreted protein, *Ixodes scapularis*	2.40 × 10^19^	2.07 × 10^2^	42.30%	1.97 × 10^9^
	Cluster-12057.0	Q202J4, Dermonecrotic toxin SPH, *Ixodes scapularis*	9.60 × 10^36^	1.94 × 10^2^	44.90%	1.44 × 10^9^
	Cluster-17602.12808	A0A182TMR5, Cuticular protein, *Anopheles melas*	2.00 × 10^15^	3.33 × 10^2^	29.70%	2.08 × 10^9^

iBAQ: intensity-based absolute quantification.

## Data Availability

The original dataset generated and analyzed during the current study is included in the Appendix A.

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
