# Peer review of "Egg Protein Compositions over Embryonic Development in Haemaphysalis hystricis Ticks"

_animals, 2024, doi:10.3390/ani14233466_

Round 1
Reviewer 1 Report
Comments and Suggestions for Authors
The group investigated and identified tick proteins from the egg stage( during days 0, 7, 14 and 21). They identified a broad range of proteins and this is important because it will in the long run help in narrowing down targets for vaccine development.
the way the work was presented was great, it was easy to follow and there were no grammatical errors detected.
My one concern is Figure 3. It would be awesome to find another way to represent that data to make it very clear to the reader. ie a bar chart or some other type of graph
Line 99 is there a reason why 350mg was used in the study any justification for that
Author Response
|
Comments 1: [The group investigated and identified tick proteins from the egg stage( during days 0, 7, 14 and 21). They identified a broad range of proteins and this is important because it will in the long run help in narrowing down targets for vaccine development. the way the work was presented was great, it was easy to follow and there were no grammatical errors detected.] Response 1: Thank you for your recognition of this study.
Comments 2: [My one concern is Figure 3. It would be awesome to find another way to represent that data to make it very clear to the reader. ie a bar chart or some other type of graph]
|
|
Response 2: Thank you for this your insightful recommendations. I’ve switched to a bar chart and hope this makes the information clearer for the reader
|
|
Comments 3: [Line 99 is there a reason why 350mg was used in the study any justification for that] |
|
Response 3: We collected hundreds of ticks, from which the largest were selected and weighed. Ticks weighing around 350 mg were considered to be engorged females. Based on our experience, engorged females will have a higher spawning rate.
|
Reviewer 2 Report
Comments and Suggestions for Authors
The manuscript is, in general, well written. Very few problems with English language. Overall, in contrast with previous studies where only one or a few proteins are studied, the manuscript is very innovative as it gives a global view of the many proteins, not transcripts, that are expressed along the embryonic development of Haemaphysalis hystricis ticks. This view presents more than a a hundred types of enzymes, transporters, immunity-related molecules, etc, and the authors propose functions of groups or individual molecules during the embryogenesis of the tick.
- Lines 214 -215, the authors mention “14 days pos-hatching”. This must be wrong as the authors work only with embryonic stages.
- Lines 268 – 270, authors mention 108 transcripts, but when adding “transporters (20), enzymes (28), immunity and antimicrobial-related proteins (7), proteinase inhibitors (20), cytoskeletal proteins (8), heat shock proteins (3), secreted proteins (9), and uncharacterized proteins (16)” equals 111 proteins
- Line 374: aspartic proteinase on day 21 post-hatching,
- Line 413, authors mention day “20”, is it 21?
- Lines 578 – 593: 3.15. Kunitz domain-containing protein: the authors show low sequence activity between R. microplus and H. hystricis and a putative anticoagulant function, which wanes as the egg matures. Thus their discussion is not clear about what possible role may have these proteins in during egg development or a hatching larvae
- Conclusions: The authors present a view of many proteins and their variations along the development and maturation of eggs from Haemaphysalis hystricis. They also propose that knowledge of the different proteins can be used in designing vaccines against the egg-stage specific proteins. One point the author fail to make, is how this knowledge would be applied in vaccine development. How is that eggs being laid on the field would be targets to effective vaccines.
- Authors fail to remind the audience the average period of incubation of eggs in the field. And go exclusively to 21 days, which in the field may be variable due to climate conditions. Notwithstanding the importance of the acquired knowledge, there is no mention as to how a vaccine towards egg proteins can be applied either in animals in the field or in stalls.
Again, the acquired knowledge justifies by itself the study, yet the lack of indication as to how knowing these proteins can be applied in the field to control tick populations.
In general the quality of written English and composition are adequate for publication.
There are a few errors in terms of expressing post-hatching with post incubation, but they have been annotated in the previous comments
Author Response
|
Comments 1: [The manuscript is, in general, well written. Very few problems with English language. Overall, in contrast with previous studies where only one or a few proteins are studied, the manuscript is very innovative as it gives a global view of the many proteins, not transcripts, that are expressed along the embryonic development of Haemaphysalis hystricis ticks. This view presents more than a a hundred types of enzymes, transporters, immunity-related molecules, etc, and the authors propose functions of groups or individual molecules during the embryogenesis of the tick.] Response 1: Thank you for your recognition of this study.
Comments 2: [Lines 214 -215, the authors mention “14 days pos-hatching”. This must be wrong as the authors work only with embryonic stages.] |
|
Response 2: Thank you for pointing out this mistake. We agree with your comment and have corrected it, changing "14 days post-hatching" to "14 days of incubation." Please refer to lines 217-218 in the tracked version.
|
|
Comments 3: [Lines 268 – 270, authors mention 108 transcripts, but when adding “transporters (20), enzymes (28), immunity and antimicrobial-related proteins (7), proteinase inhibitors (20), cytoskeletal proteins (8), heat shock proteins (3), secreted proteins (9), and uncharacterized proteins (16)” equals 111 proteins] |
|
Response 3: Sorry, we made a mistake for the number of cytoskeletal proteins, which should be five, I’ve changed it that can be found in lines 273-274 of tracking version.
Comments 4: [Line 374: aspartic proteinase on day 21 post-hatching,] Response 4: Sorry for this again and have corrected it.
Comments 5: [Line 413, authors mention day “20”, is it 21?,] Response 5: No, it refers to a study by Freitas et al. (2007), which found that GSPs increase during egg development, peaking on day 20. This is consistent with our findings.
Comments 6: [Lines 578 – 593: 3.15. Kunitz domain-containing protein: the authors show low sequence activity between R. microplus and H. hystricis and a putative anticoagulant function, which wanes as the egg matures. Thus their discussion is not clear about what possible role may have these proteins in during egg development or a hatching larvae,] Response 6: Yes, the low sequence identities suggest two possibilities: 1) they may not be Kunitz proteins, or 2) the sequence identities within the Kunitz family are generally low. Many studies have shown that Kunitz proteins possess anticoagulant functions, which leads us to hypothesize that Kunitz proteins could be involved in anticoagulation. However, no studies have reported Kunitz proteins being involved in anticoagulation specifically in tick eggs. If they involved in anticoagulant, the reason for the decrease in Kunitz abundance remains unclear.
Comments 7: [Conclusions: The authors present a view of many proteins and their variations along the development and maturation of eggs from Haemaphysalis hystricis. They also propose that knowledge of the different proteins can be used in designing vaccines against the egg-stage specific proteins. One point the author fail to make, is how this knowledge would be applied in vaccine development. How is that eggs being laid on the field would be targets to effective vaccines.] Response 7:. Thank you for your constructive feedback. We agree that our conclusion section could benefit from a more detailed explanation of how the identified proteins could be applied in vaccine development. In the revised manuscript, we will address this point more explicitly. Based on the current study, we aim to identify tick egg proteins and investigate their expression levels during egg development. And then we can select candidate proteins based on their changes and importance during egg development for further investigation. For example, we can isolate their transcripts, obtain full-length mRNA, and then perform RNA interference. Additionally, we can express the proteins and assess their functions in vitro.
Comments 8: [Authors fail to remind the audience the average period of incubation of eggs in the field. And go exclusively to 21 days, which in the field may be variable due to climate conditions. Notwithstanding the importance of the acquired knowledge, there is no mention as to how a vaccine towards egg proteins can be applied either in animals in the field or in stalls. Again, the acquired knowledge justifies by itself the study, yet the lack of indication as to how knowing these proteins can be applied in the field to control tick populations.] Response 8: Thank you for pointing that out. Yes, the incubation period will vary in the field. We have cited studies related to the incubation period of Haemaphysalis hystricis eggs, and remind that the incubation period may vary depending on climate conditions in the field The proteins identified in our study will serve as a reference for selecting candidates for further research, including RNAi, in vitro protein expression studies, protein function experiment, etc. More research is needed to fully understand their functions, and once their roles are clarified, these proteins can potentially be applied in the field to control tick populations.
|
Reviewer 3 Report
Comments and Suggestions for Authors
Dear authors
The manuscript is an important contribution and deserves to be published with minor revisions.
I asked some questions in Materials and Methods because I felt there were no citations and also in Results and discussion.

Author Response
|
Comments 1: [The manuscript is an important contribution and deserves to be published with minor revisions. I asked some questions in Materials and Methods because I felt there were no citations and also in Results and discussion.] Response 1: Thank you for recognizing the value of this study. Please find the detailed responses below, along with the corresponding corrections marked in track changes in the re-submitted files.
Comments 2: [related to the italic for species in lines 48, 61, 89, 190, 234, 310, ]
|
|
Response 2: Thank you for the detailed reminder, I’ve corrected it.
|
|
Comments 3: [line 54: cite author] |
|
Response 3: Thank you for pointing out this. I have already cited one.
Comments 4: [line 96: please cite the taxonomic key and who published.] Response 4: Thank you for pointing out this. I have cited a study related to the morphological identification of Haemaphysalis hystricis, which is a study published in Chinese.
Comments 5: [lines 106-108: Whose protocol is it?It's basic, but you need to cite who described the technique] Response 5: Thank you for pointing out this. I have cited a study related to tick egg dewaxing process.
Comments 6: [lines 113-124: Whose protocol is it?] Response 6: We extracted the protein using the protocol provided with the protein extraction kit purchased from Sigma, St. Louis, MI, USA. I’ve added this information to line 121.
Comments 7: [line 223: include 2015 or number 28. Anyway they demonstrated this to H. longicornis. Do you know if there is proteomic study for females of H. hytricis? comment it.] Response 7: Apology for this, we didn’t find any published proteomic study for females of H. hytricis. Seems that this is the first proteomic report for females of H. hytricis. We would appreciate it if anyone could point to any relevant studies on this topic.
Comments 8: [line 326: Were lipocalins present only in the eggs on day 1? What happened on days 7, 14 and 21?] Response 8: It is unfortunate that we did not select lipocalin for PRM analysis, so we are unable to assess its changes during egg development. Further studies are needed to explore the changes in lipocalin during egg development.
Comments 8: [line 332: limited knowledge - please comment other studies about egg protein expression.] Response 8: There are limited reports on egg protein expression, with only one study related to lipocalins in tick eggs, as mentioned in lines 343-347. Therefore, the functions of these proteins in tick eggs remain unknown. We would appreciate it if you could provide any additional studies related to lipocalins in tick eggs.
Comments 9: [line 453: Which tick cell lines? Please comment.] Response 9: Thank you for pointing out this. It is Ixodes scapularis.
Comments 10: [lines 363&459: missing “.”.] Response 10: Thank you for pointing out this. Added.
|